# LIPNEXT: SCALING UP LIPSCHITZ-BASED CERTIFIED ROBUSTNESS TO BILLION-PARAMETER MODELS

**Kai Hu, Haoqi Hu, Matt Fredrikson**
Carnegie Mellon University
`kaihu@cs.cmu.edu`

## ABSTRACT

Lipschitz-based certification offers efficient, deterministic robustness guarantees but has struggled to scale in model size, training efficiency, and ImageNet performance. We introduce *LipNeXt*, the first *constraint-free* and *convolution-free* 1-Lipschitz architecture for certified robustness. LipNeXt is built using two techniques: (1) a manifold optimization procedure that updates parameters directly on the orthogonal manifold and (2) a *Spatial Shift Module* to model spatial pattern without convolutions. The full network uses orthogonal projections, spatial shifts, a simple 1-Lipschitz $\beta$-Abs nonlinearity, and $L_2$ spatial pooling to maintain tight Lipschitz control while enabling expressive feature mixing. Across CIFAR-10/100 and Tiny-ImageNet, LipNeXt achieves state-of-the-art clean and certified robust accuracy (CRA), and on ImageNet it scales to 1–2B large models, improving CRA over prior Lipschitz models (e.g., up to $+8\%$ at $\varepsilon{=}1$) while retaining efficient, stable low-precision training. These results demonstrate that Lipschitz-based certification can benefit from modern scaling trends without sacrificing determinism or efficiency.

## 1 INTRODUCTION

Adversarial robustness represents a fundamental challenge in machine learning (Szegedy et al., 2014). Numerous defense mechanisms have been developed to enhance model robustness against adversarial attacks (Gong et al., 2021; Kundu et al., 2021; Poursaeed et al., 2021; Liu et al., 2021; Pang et al., 2021). However, these approaches are predominantly empirical defenses that cannot provide formal guarantees of robust predictions. Consequently, models deemed robust under current evaluation protocols may remain vulnerable to more sophisticated attack strategies as they emerge.

This limitation is particularly concerning for safety-critical applications such as autonomous driving (Huang et al., 2025), medical image processing (Laousy et al., 2023), and malware classification (Saha et al., 2024), where failures can have severe consequences. To address this challenge, certified robustness has emerged as a promising research direction. Certifying the robustness of a test case requires a mathematical guarantee that the model's outputs remain unchanged within a predefined $\ell_p$-norm ball of radius $\varepsilon$ around the input. The performance of these certification methods is typically measured by the *certified robust accuracy* (CRA), which quantifies the proportion of correctly predicted inputs that are also provably robust within a specified radius.

Research on robustness certification largely follows two methodological strands. The first, *randomized smoothing* (RS) (Cohen et al., 2019b; Yang et al., 2021; Jeong et al., 2021; Carlini et al., 2022), provides *probabilistic* guarantees by averaging a classifier's predictions under additive noise. The second exploits the Lipschitz properties of neural networks to yield *deterministic* (worst-case) certificates (Huang et al., 2021; Araujo et al., 2023; Hu et al., 2024). In this work, we focus on advancing the latter direction; a detailed comparison between the two appears in Appendix A.

Despite their theoretical appeal, Lipschitz-based certification methods have struggled to scale in practice. A central critique is the weak performance on large-scale benchmarks: models often underfit even small-sized datasets such as CIFAR-100 and degrade markedly on ImageNet (Hu et al., 2023). Most systems still rely on *small*, VGG-style (Simonyan & Zisserman, 2014) architectures with $\leq$32M parameters. Although recent work has explored deeper/larger architectures for certified robustness (Hu et al., 2023; Araujo et al., 2023), the gains plateau quickly as model size increases.

Examining existing work, we find that orthogonal matrices are fundamental to building 1-Lipschitz networks because they enable tight Lipschitz bounds. However, they are also a major bottleneck that prevents Lipschitz-based certification from scaling. Existing methods either explicitly re-parameterize orthogonal matrices or implicitly re-parameterize Lipschitz-bounded operations to learn near-orthogonal matrices; both introduce substantial computational overhead (see Section 2.1). To address this issue, we propose directly optimizing orthogonal matrices on the orthogonal manifold. Although orthogonal manifold optimization is a mature technique, to our knowledge it has not been exploited for certification. We further observe that in the large-model regime, where learning rates are small, the matrix exponential can be accurately and efficiently approximated. Combining these ideas, we reduce the additional per-update overhead to at most five matrix multiplications.

A natural path to scalability is transformers (Vaswani et al., 2017), which scale to billion-parameter models and exhibit emergent abilities across a wide range of tasks (Wei et al., 2022). However, attention lacks straightforward Lipschitz control. Fortunately, ConvNeXt (Liu et al., 2022) and MetaFormer (Yu et al., 2022b) suggest that Lipschitz-bounded architectures can benefit from transformer-era design choices. Motivated by the simple token-mixing mechanisms in these modern architectures (Yu et al., 2022b), we design a convolution-free shifting module paired with positional encoding that uses simple shift operations to model spatial relations as the building block of our model. Compared to prior work using heavy FFT-based convolution designs (Trockman & Kolter, 2021; Lai et al., 2025) or power-iteration-based Lipschitz regularization (Leino et al., 2021; Hu et al., 2023), the shifting module adds minimal computational cost while preserving the capacity to model spatial patterns.

Combining the aforementioned techniques, we propose **LipNeXt**, the first convolution-free and constraint-free architecture for certified robustness. Benefiting from this design, we scale to billion-parameter models and observe non-saturating gains with increasing model size. Our experiments show that LipNeXt outperforms state-of-the-art methods in both clean accuracy and certified robustness across CIFAR-10, CIFAR-100, Tiny-ImageNet, and ImageNet, highlighting the effectiveness of our approach for scalable, provably robust deep learning.

**Contributions.** (i) A constraint-free manifold optimization scheme for learning orthogonal parameters at scale with efficient exponential approximations and stabilization; (ii) a theoretically motivated, convolution-free Spatial Shift Module that preserves Lipschitz control while enabling spatial mixing; and (iii) state-of-the-art CRA and clean accuracy on standard benchmarks, with successful scaling to billion-parameter models.

## 2 PRELIMINARY AND RELATED WORK

### 2.1 PRELIMINARIES

**Lipschitz-based Method for Certified Robustness**   Consider a neural network $f : \mathbb{R}^d \to \mathbb{R}$ and its corresponding binary classifier $F(x) = \text{sign}(f(x))$. We say that classifier $F$ is $\varepsilon$-**locally robust** at point $x$ if for all perturbations $x'$ satisfying $\|x - x'\|_p \leq \varepsilon$, we have $F(x) = F(x')$. **Certified robustness** represents a stronger condition where the robustness property cannot only be empirically verified but also mathematically guaranteed. In this work, we only focus on $p = 2$, i.e., certified robustness under $\ell_2$ norm.

Given an upper bound $K$ on the Lipschitz constant of $f$, we can certify that $F$ is locally robust at $x$ with a guaranteed robustness radius of $|f(x)|/K$. This certification follows from the Lipschitz property, which bounds the maximum change in $f(x)$ within the $\varepsilon$-ball around $x$.

For multi-class classification with $N$ classes, we decompose the problem into $N - 1$ binary classification tasks using a one-vs-rest approach. The certified robustness radius of the $N$-class classifier is then defined as the minimum certified radius across all constituent binary classifiers, thereby ensuring robustness for the prediction.

To compute Lipschitz bounds for deep networks, we leverage the compositional property that the Lipschitz constant of a composite function satisfies $\text{Lip}(f \circ g) \leq \text{Lip}(f)\text{Lip}(g)$, where $(f \circ g)(x) = f(g(x))$. Consequently, for a feed-forward neural network, an upper bound on the overall Lipschitz constant can be efficiently computed as the product of the Lipschitz constants of all constituent

layers. However, this bound can be very loose, leading to an extremely small certified radius. Therefore, designing a network architecture for which a tight Lipschitz bound can be computed is crucial to achieving satisfactory certified performance.

**Orthogonal Manifold Optimization**    The orthogonal manifold is defined as the set of all orthogonal matrices: $\mathcal{M}_d = \{\mathbf{X} \in \mathbb{R}^{d \times d} \mid \mathbf{X}^\top \mathbf{X} = \mathbf{I}_d\}$, where $\mathbf{I}_n$ denotes the $d \times d$ identity matrix. Optimization problems on the orthogonal manifold can be formulated as:

$$\min_{\mathbf{X} \in \mathcal{M}_d} f(\mathbf{X}). \tag{1}$$

A standard Euclidean gradient descent update of the form $\mathbf{X}_{k+1} \leftarrow \mathbf{X}_k - \eta \nabla f(\mathbf{X}_k)$ with step size $\eta > 0$ will generally cause $\mathbf{X}_{k+1}$ to leave the manifold, since this update does not preserve orthogonality constraints. Manifold optimization first projects the Euclidean gradient onto the tangent space of the manifold, thereby removing the component of the gradient that is orthogonal to the manifold. The Riemannian gradient (projected gradient) is given by:

$$\mathrm{grad}\, f(\mathbf{X}) = \nabla f(\mathbf{X}) - \mathbf{X}\, \mathrm{sym}(\mathbf{X}^\top \nabla f(\mathbf{X})), \tag{2}$$

where $\mathrm{sym}(\mathbf{A}) = (\mathbf{A} + \mathbf{A}^\top)/2$ denotes the symmetric part of matrix $\mathbf{A}$. Once the Riemannian gradient is computed, **Exponential Map** is employed to update $\mathbf{X}_{k+1}$ while preserving orthogonality:

$$\mathbf{X}_{k+1} \leftarrow \mathbf{X}_k \exp\left\{-\frac{\eta}{2} \mathrm{skew}(\mathbf{X}_k^\top \mathrm{grad}\, f(\mathbf{X}_k))\right\}, \tag{3}$$

where $\mathrm{skew}(\mathbf{A}) = (\mathbf{A} - \mathbf{A}^\top)/2$, and $\exp(\mathbf{A})$ denotes the matrix exponential. Orthogonality is preserved since $\exp(\mathrm{skew}(\mathbf{A}))$ is orthogonal for any matrix $\mathbf{A}$. Bécigneul & Ganea (2018) extended this optimization to adaptive optimizers like Adam, and algorithm 2 provides an example for Manifold Adam Optimizer.

## 2.2    RELATED WORK

**Re-parameterization for Lipschitz-based Certified Robustness**    The Lipschitz upper bound discussed in Section 2.1 can be tight when all weights in the feed-forward network are orthogonal or near-orthogonal. Many methods have been proposed to parameterize weights as orthogonal matrices to enhance Lipschitz-based certification performance. Singla & Feizi (2021) employed the matrix exponential of a skew-symmetric matrix to construct orthogonal (1-Lipschitz) linear layers and extended this approach to convolutions via Taylor expansion. Trockman & Kolter (2021) introduced the Cayley transform, $\mathrm{Cayley}(\mathbf{A}) = (\mathbf{I} - \mathbf{A})^{-1}(\mathbf{I} + \mathbf{A})$, which yields an orthogonal matrix when $\mathbf{A}$ is skew-symmetric. To apply this method to convolutions, the weights are transformed into the frequency domain via FFT before applying the Cayley transform. Xu et al. (2022) proposed LOT-Orth, defined as $(\mathbf{A}\mathbf{A}^\top)^{-1/2}\mathbf{A}$. To mitigate the computational overhead introduced by the matrix square root in LOT-Orth, Hu et al. (2024) proposed Cholesky decomposition with $\mathrm{Cholesky\text{-}Orth}(\mathbf{A}) = \mathrm{Cholesky}(\mathbf{A}\mathbf{A}^\top)^{-1}\mathbf{A}$.

Alternative approaches focus on near-orthogonal architectures. Prach & Lampert (2022) introduced the AOL layer: $f(x, \mathbf{A}) = \mathbf{A}\mathrm{diag}(\sum_j |\mathbf{A}^\top \mathbf{A}|_{ij})^{-1}x$. Meunier et al. (2022) proposed the 1-Lipschitz CPL layer defined as $f(x, \mathbf{A}) = x - 2\mathbf{A}^\top \sigma(\mathbf{A}x)/|\mathbf{A}|_2^2$. Araujo et al. (2023) introduced the SLL layer given by $f(x, \mathbf{A}, q) = x - 2\mathbf{A}\mathrm{diag}(\sum_j |\mathbf{A}^\top \mathbf{A}|_{ij} \cdot q_j/q_i)^{-1}\sigma(\mathbf{A}^\top x)$. While the AOL, CPL, and SLL layers may not be orthogonal at initialization, experimental evidence shows that these layers converge to nearly orthogonality at the end of Lipschitz-based training.

**Efficient Manifold Optimization for Orthogonality**    While the matrix exponential operation ensures orthogonality in manifold optimization (Section 2.1), its computation is computationally prohibitive. Several methods have proposed efficient pseudo-geodesic retractions as alternatives to exponential mapping. Projection-based approaches by Absil & Malick (2012); Gawlik & Leok (2018) map gradients back to the orthogonal manifold but rely on computationally expensive SVD operations. Other methods utilize the closed-form Cayley transform (Jiang & Dai, 2015; Zhu, 2017), yet require costly matrix inversions. Wen & Yin (2013) reduced the computational cost of the Cayley transform by imposing the restrictive assumption that for matrix dimensions $n \times p$ if $2p \ll n$. However, this algorithm becomes inefficient when $2p \geq n$, which represents the majority of cases in deep neural networks. Shustin & Avron (2024) employ randomized sketching methods to handle

large update matrices efficiently. Ablin & Peyré (2022) proposed a landing algorithm that avoids costly matrix exponential retractions by using an efficient potential energy-based update scheme that progressively pulls iterations onto the manifold.

## 3 METHOD

Our proposed method, LipNeXt, is built upon two core technical innovations designed for scalable certified robustness: a constraint-free optimization technique for learning orthogonal matrices and a novel convolution-free spatial mixing operator. In this section, we first detail our manifold optimization approach, which enables efficient and stable training of orthogonal parameters directly on the Stiefel manifold. We then introduce the **Spatial Shift Module**, providing theoretical justification for its design as the unique norm-preserving operator within the family of depth-wise convolutions. Finally, we describe the overall LipNeXt architecture, which integrates these components into a scalable and effective model.

### 3.1 CONSTRAINT-FREE LEARNING OF ORTHOGONAL MATRICES ON THE MANIFOLD

As discussed in Section 2.2, prior works use re-parameterization to learn orthogonal matrices explicitly or implicitly. We name these as *constrained* approaches, as the optimization variables do not reside directly on the orthogonal manifold $\mathcal{M}_d$. In contrast, we adopt a *constraint-free* manifold optimization perspective, where parameters are updated directly on the manifold, inherently preserving orthogonality. Section 2.1 introduces the details of orthogonal manifold optimization.

A significant bottleneck in manifold optimization is the computational cost of the matrix exponential, $\exp(\cdot)$ in Equation 3. Section 2.2 discussed solutions from prior work to address this issue.

However, we observe that there exists a very simple solution in the context of training neural networks. As the model size increases, the optimal learning rate for optimizers like Adam needs to decrease, (e.g., $10^{-3}$). Thus the exponential component in Equation 3 often has a smaller Frobenius norm (because multiplied by the learning rate $\eta$). We propose approximating the exponential with a norm-adaptive Taylor series truncation. For a given skew-symmetric update matrix $A$, we define:

$$\text{FastExp}(A) = \begin{cases} I + A + \frac{1}{2}A^2, & \text{if } \|A\|_F < 0.05, \\ I + A + \frac{1}{2}A^2 + \frac{1}{6}A^3, & \text{if } 0.05 \leq \|A\|_F < 0.25, \\ I + A + \frac{1}{2}A^2 + \frac{1}{6}A^3 + \frac{1}{24}A^4, & \text{if } 0.25 \leq \|A\|_F < 1, \\ \exp(A), & \text{if } \|A\|_F \geq 1. \end{cases} \tag{4}$$

Here the hyper-parameters 0.05, 0.25 and 1 are chosen empirically to balance accuracy and stability. One issue of $\text{FastExp}(\cdot)$ is the truncation introduces small errors that violate exact orthogonality, which can accumulate and destabilize training over time. To address this issue, we introduce two stabilization techniques and mark them in Algorithm 1.

First, to control the accumulation of numerical errors from the truncated series, we perform a **periodic polar retraction**. At the end of each epoch, we project the matrix $X$ back to the manifold. This is achieved by computing the Singular Value Decomposition $X = U\Sigma V^\top$ and resetting $X \leftarrow UV^\top$, which finds the closest orthogonal matrix to $X$ in the Frobenius norm. Although SVD is also expensive, we only need to perform it once per epoch. This step is detailed in lines 20–22 (red text) of Algorithm 1.

Second, we adapt the Lookahead optimizer wrapper (Zhang et al., 2019) to the manifold setting. In a nutshell, Lookahead optimizer wrapper updates the optimized parameters with an interpolation between the correct weight and the weight $K$ steps earlier every $K$ steps:

$$\text{If } (t+1) \bmod K = 0, \quad X_t \leftarrow 0.5X_t + 0.5X_{t-K}$$

However direct interpolation of orthogonal matrices breaks orthogonality. Applying the polar retraction after interpolation is also expensive if applied every $K$ steps.

We instead interpolate the *skew-symmetric updates* in the tangent space. Suppose the learning rate is small, and the exponential component of update ($\Delta_j$) at every step is small:

$$X_t = X_{t-1}\exp(\Delta_t) = X_{t-K}\prod_{j=t-K+1}^{t}\exp(\Delta_j) \approx X_{t-K}\exp(\sum_{j=t-K+1}^{t}\Delta_j). \qquad (5)$$

Thus we can approximate $0.5X_t + 0.5X_{t-K}$ as $X_{t-K}\exp(\frac{1}{2}\sum_{j=t-k+1}^{t}\Delta_j)$. To understand this, starting at $x_{t-K}$, we update the parameters $K$ steps and collect the update trajectories $\{\Delta_j\}_{t-K+1}^{t}$, then we go back to $X_{t-K}$, and update $X_{t-K}$ using the half of all $K$ trajectories. Lines 12∼28 (blue text) of Algorithm 1 reflect this method.

---

**Algorithm 1** Stabilized Manifold Adam Optimizer with FastExp

---

1: **Input:** learning rate $\eta$ (on the order of $10^{-3}$), momentum coefficients $\beta_1$ and $\beta_2$. Number of steps to perform a Lookahead update $K$, and number of steps in one epoch $N$.
2: **Goal:** Minimize $f(X)$ on the orthogonal manifold beginning at orthonormal weight $X \in \mathcal{M}_d$.
3: Set the fast weight $X_0 \leftarrow X$ and slow weight $X_{\text{slow}} \leftarrow X$
4: Set the Lookahead updating buffer $B_0 \leftarrow \mathbf{0}^{d\times d}$.
5: Set the first and second moment $m_0 = \mathbf{0}^{d\times d}, v_0 = \mathbf{1}^{d\times d}/d$.
6: **for** step $t$ in $1\cdots, T$ **do**
7:     Compute the Euclidean gradient $\nabla f(X_{t-1})$.
8:     Compute the projected gradient $\text{grad}\, f(X_{t-1})$ using $\nabla f(X_{t-1})$ and $X_{t-1}$ by Equation 2.
9:     Update the first order moment: $m_t \leftarrow \beta_1 m_{t-1} + (1-\beta_1)\text{grad}\, f((X_{t-1})$.
10:     Update the second order moment: $v_t \leftarrow \beta_2 v_{t-1i} + (1-\beta_2)\text{grad}\, f(X_{t-1}) * \text{grad}\, f(X_{t-1})$.
11:     Compute the exponential component $\Delta_t \leftarrow -\eta \cdot m_t/v_t$.
12:     Update the Lookahead updating buffer $B_t \leftarrow B_t + \Delta_t$.
13:     **if** $(t+1) \bmod K \neq 0$ **then**
14:         Update the fast weight: $X_t \leftarrow X_{t-1}\text{FastExp}(\Delta_t)$.
15:     **else**
16:         Update the slow weight: $X_{\text{slow}} \leftarrow X_{\text{slow}}\text{FastExp}(B_t/2)$.
17:         Update the fast weight with the slow weight: $X_t \leftarrow X_{\text{slow}}$.
18:         Unset the updating buffer $B_t \leftarrow \mathbf{0}^{n\times n}$.
19:     **end if**
20:     **if** $(t+1) \bmod N = 0$ **then**
21:         Perform SVD on the fast weight: $X_t = U\Sigma V^\top$.
22:         Force orthogonalization: $X_t \leftarrow UV^\top, X_{\text{slow}} \leftarrow X_t$.
23:     **end if**
24: **end for**
25: **Output:** $X_{\text{slow}}$.

---

**Relation to skew re-parameterization** Skew re-parameterization constructs orthogonal matrices via the matrix exponential of a skew-symmetric argument, i.e., $\exp(S)$ with $S = X - X^\top$. Building on this idea, Singla & Feizi (2021) applies a truncated Taylor expansion of the matrix exponential to implement orthogonal convolutions. In contrast, our manifold optimization exploits the regime of *small* per-step updates, allowing the exponential map to be accurately approximated with lightweight computations. As a result, we retain orthogonality with significantly lower cost. By comparison, skew re-parameterization cannot leverage small-update structure in the same way, and SOC typically requires a larger truncation order to reliably preserve orthogonality, increasing computational overhead and potential numerical error.

### 3.2 SPATIAL SHIFT MODULE: A CONVOLUTION-FREE DESIGN

Depthwise separable convolutions have demonstrated that effective spatial pattern learning requires neither extensive parameterization nor computational overhead (Howard et al., 2017; Tan & Le, 2019; Liu et al., 2022). MetaFormer (Yu et al., 2022b) further showed that parameter-free operations can replace depthwise convolutions while maintaining performance. Building on this trajectory

toward parameter efficiency, we introduce a convolution-free **Spatial Shift Module** tailored for certified robustness.

**1D formulation.** For input sequence $X = [x_1, \ldots, x_n] \in \mathbb{R}^{d \times n}$ of length $n$ with feature dimension $d$, we partition each token's features into three parts: $a_i \in \mathbb{R}^{\alpha d}$, $b_i \in \mathbb{R}^{\alpha d}$, and $c_i \in \mathbb{R}^{(1-2\alpha)d}$. The spatial shift operation $\mathcal{S}(X)$ applies circular shifts to the first two partitions:

$$X = \begin{bmatrix} a_1 & a_2 & \cdots & a_n \\ b_1 & b_2 & \cdots & b_n \\ c_1 & c_2 & \cdots & c_n \end{bmatrix}, \quad \mathcal{S}(X) = \begin{bmatrix} a_n & a_1 & \cdots & a_{n-2} & a_{n-1} \\ b_2 & b_3 & \cdots & b_n & b_1 \\ c_1 & c_2 & \cdots & c_{n-1} & c_n \end{bmatrix} \quad (6)$$

This shifts the first partition right by one position and the second partition left by one position, mixing adjacent token information without parameters. For 2D data, we extend to five partitions enabling shifts along horizontal and vertical axes. Empirically, shift ratios $\alpha \in \{1/8, 1/16\}$ yield optimal results.

**Theoretical justification.** Our design is motivated by norm-preservation requirements in certified robustness. The following theorem establishes fundamental constraints on Lipschitz-preserving convolutions:

**Theorem 1** *Let $X \in \mathbb{R}^{H \times W}$ be a single-channel tensor and $f_K$ be spatial convolution with kernel $K \in \mathbb{R}^{k \times k}$, unit stride, and circular padding (crucially relies on circular padding and does not hold under zero-padding). The operator $f_K$ is norm-preserving (tight 1-Lipschitz isometric):*

$$\|f_K(X) - f_K(Y)\|_F = \|X - Y\|_F, \quad \forall X, Y \in \mathbb{R}^{H \times W}$$

*if and only if kernel $K$ contains exactly one non-zero element with value $\pm 1$.*

Theorem 1 reveals that norm-preserving depthwise convolutions reduce to spatial shifts. Our module directly implements this principle. Proof of Theorem 1 is in Appendix B.3.

**Circular padding and positional encoding.** The strict norm preservation guaranteed by Theorem 1 necessitates the use of circular padding. Nevertheless, in related applications such as FFT-based convolution, zero-padding is a common alternative (Xu et al., 2022; Lai et al., 2025). The rationale for this deviation is to avoid the artificial mixing of boundary features (e.g., $a_n$ with $b_2$) that can occur with circular padding.

$$\mathcal{S}_{\text{zero-pad}}(X) = \begin{bmatrix} \mathbf{0} & a_1 & \cdots & a_{n-2} & a_{n-1} \\ b_2 & b_3 & \cdots & b_n & \mathbf{0} \\ c_1 & c_2 & \cdots & c_{n-1} & c_n \end{bmatrix} \quad (7)$$

We also observe $\mathcal{S}_{\text{zero-pad}}$ outperforms $\mathcal{S}$ in our convolutional-free architecture. However, we would explain this with another hypothesis that zero-padding introduces position information to the model (Islam et al., 2024). Under this hypothesis, we can address the issue by add explicit positional encoding to the input and keep the circular padding. Our experiments in Section 4 verifies that this hypothesis is correct: by using circular padding $\mathcal{S}$ and explicit positional embedding, the model achieving superior performance because of norm preservation guarantees.

### 3.3 The LipNeXt Architecture

We now define the LipNeXt architecture, which integrates our manifold optimizer and Spatial Shift Module. The core of the architecture is the LipNeXt block.

Let $X \in \mathbb{R}^{H \times W \times C}$ be the input tensor. First, we add a **learnable positional embedding** $p \in \mathbb{R}^{H \times W \times 1}$, which is broadcast across the channel dimension: $X' = X + p$. Next, we apply the core mixing operation. We use an orthogonal matrix $R \in \mathcal{M}_C$ to project the channel-wise features, apply the 2D Spatial Shift Module $\mathcal{S}$, and then project back using $R^\top$:

$$Y = R^\top \mathcal{S}(RX') \in \mathbb{R}^{H \times W \times C}. \quad (8)$$

The projections $R$ and $R^\top$ ensure that the shift operation is not always applied to the same fixed subset of channels, enabling comprehensive feature mixing across all channels over successive blocks. Without $\mathcal{S}$, this operation reduces to an identity mapping $Y = R^\top R X' = X'$. To learn more complex transformations, we apply another orthogonal matrix $M \in \mathcal{M}_C$ and an activation $\sigma$:

$$Z = \sigma(MY + b) = \sigma(MR^\top \mathcal{S}(R(X + p)) + b). \tag{9}$$

For the activation, we propose $\beta$-**Abs**, a flexible and GPU-friendly 1-Lipschitz function. For an input vector $\boldsymbol{x} \in \mathbb{R}^d$, it is defined channel-wise as:

$$[\beta\text{-Abs}(\boldsymbol{x})]_i = \begin{cases} |x_i|, & \text{if } i \le \beta d \\ x_i, & \text{otherwise.} \end{cases} \tag{10}$$

$\beta$-Abs is a simple yet powerful non-linearity. We can show that the commonly used MinMax activation can be expressed by $\beta$-Abs if $\beta = 0.5$:

$$\exists R \in \mathcal{M}_{2d}, \forall x \in \mathcal{R}^{2d} \quad \text{MinMax}(x) = R^\top \beta\text{-Abs}(Rx). \tag{11}$$

The hyperparameter $\beta \in [0, 1]$ controls the degree of non-linearity. We defer further discussion to the Appendix B.1. A PyTorch-like code of the LipNeXt block is given at Appendix B.2.

The overall LipNeXt architecture adapts the macro-structure of LiResNet (Hu et al., 2023). We replace the original LiResNet blocks with our LipNeXt blocks and substitute the Neck module with a simple pooling layer, as our manifold optimizer is designed for square matrices. Following the philosophy of Vision Transformers (Dosovitskiy et al., 2020), we rely on the scalability of the backbone to learn effective features. To ensure the entire network is 1-Lipschitz, we use an **L2 Spatial Pool**, which computes the L2 norm over the spatial dimensions for each channel. For an input $X \in \mathbb{R}^{H \times W \times C}$, the output is a vector in $\mathbb{R}^C$ where the $c$-th component is:

$$[\text{L2SpatialPool}(X)]_c = \sqrt{\sum_{h=1}^{H} \sum_{w=1}^{W} X_{h,w,c}^2}. \tag{12}$$

This operation is 1-Lipschitz and serves as the final global pooling before classification.

## 4 EXPERIMENTS

In this section, we present the empirical evaluation of our approach. We first compare our method with the SOTA certified robustness methods on the widely used CIFAR10, CIFAFR100 and Tiny-ImageNet benchmark. Next we conduct scaling experiments to show the scalability of our model in terms of network depth, network with and dataset size. We use the EMMA loss (Hu et al., 2023) for our training and follow the same training receipts in LiResNet++ (Hu et al., 2024). See Code for implementation details. By default, we do not use diffusion-generated synthetic data.

**Main Results** We compare LipNeXt against SOTA baselines. Table 1 reports clean accuracy, certified robust accuracy (CRA), and parameter counts. For LiResNet (Hu et al., 2023), we follow the setting without diffusion-generated synthetic data, using the numbers reported by Lai et al. (2025). Results that leverage diffusion-generated data are presented separately in the next table. LipNeXt achieves the strongest performance on nearly all metrics; the only exception is CRA at $\varepsilon = 108/255$ on CIFAR-10, where AOL (Prach & Lampert, 2022) is higher. However, AOL attains this point by incurring a substantial drop in clean accuracy and in CRA at the other radii. Even under the similar parameter budgets, our smaller configuration L32W1024 outperforms prior work by clear margins.

**Performance with extra data** As we further scale up LipNeXt, we continue to observe improvement in clean accuracy, but a decrease in CRA. This is a known issue called robust overfitting (Rice et al., 2020; Yu et al., 2022a) due to the small dataset size. Following prior work (Hu et al., 2024; Lai et al., 2025), we add diffusion-generated synthetic data to the training set to solve this issue, and use the generated data provided by Hu et al. (2024). Table 2 shows LipNeXt can effectively leverages these synthetic datasets to enhance performance.

Table 1: Clean and Certified robust accuracy (CRA) of prior works and our LipNeXt models on CIFAR-10/100 and TinyImageNet. The best results are marked in bold.

| Datasets | Models | #Param. | Clean Acc.(%) | CRA. (%) at ($\varepsilon$) | | |
|---|---|---|---|---|---|---|
| | | | | $36/255$ | $72/255$ | $108/255$ |
| CIFAR-10 | Cayley Large (Trockman & Kolter, 2021) | 21M | 74.6 | 61.4 | 46.4 | 32.1 |
| | SOC-20 (Singla & Feizi, 2021) | 27M | 76.3 | 62.6 | 48.7 | 36.0 |
| | LOT-20 (Xu et al., 2022) | 18M | 77.1 | 64.3 | 49.5 | 36.3 |
| | CPL XL (Meunier et al., 2022) | 236M | 78.5 | 64.4 | 48.0 | 33.0 |
| | AOL Large (Prach & Lampert, 2022) | 136M | 71.6 | 64.0 | 56.4 | **49.0** |
| | SLL X-Large (Araujo et al., 2023) | 236M | 73.3 | 64.8 | 55.7 | 47.1 |
| | LiResNet (Hu et al., 2024) | 83M | 81.0 | 69.8 | 56.3 | 42.9 |
| | BRONet (Lai et al., 2025) | 68M | 81.6 | 70.6 | 57.2 | 42.5 |
| | LipNeXt L32W1024 | 64M | 81.5 | 71.2 | **59.2** | 45.9 |
| | LipNeXt L32W2048 | 256M | **85.0** | **73.2** | 58.8 | 43.3 |
| CIFAR-100 | Cayley Large (Trockman & Kolter, 2021) | 21M | 43.3 | 29.2 | 18.8 | 11.0 |
| | SOC-20 (Singla & Feizi, 2021) | 27M | 47.8 | 34.8 | 23.7 | 15.8 |
| | LOT-20 (Xu et al., 2022) | 18M | 48.8 | 35.2 | 24.3 | 16.2 |
| | CPL XL (Meunier et al., 2022) | 236M | 47.8 | 33.4 | 20.9 | 12.6 |
| | AOL Large (Prach & Lampert, 2022) | 136M | 43.7 | 33.7 | 26.3 | 20.7 |
| | SLL X-Large (Araujo et al., 2023) | 236M | 47.8 | 36.7 | 28.3 | 22.2 |
| | Sandwich (Wang & Manchester, 2023) | 26M | 46.3 | 35.3 | 26.3 | 20.3 |
| | LiResNet (Hu et al., 2024) | 83M | 53.0 | 40.2 | 28.3 | 19.2 |
| | BRONet(Lai et al., 2025) | 68M | 54.3 | 40.2 | 29.1 | 20.3 |
| | LipNeXt L32W1024 | 64M | 53.3 | 41.3 | 30.5 | 21.8 |
| | LipNeXt L32W2048 | 256M | **57.4** | **44.1** | **31.9** | **22.2** |
| Tiny-ImageNet | SLL X-Large (Araujo et al., 2023) | 1.1B | 32.1 | 23.2 | 16.8 | 12.0 |
| | Sandwich (Wang & Manchester, 2023) | 39M | 33.4 | 24.7 | 18.1 | 13.4 |
| | LiResNet[†] (Hu et al., 2024) | 83M | 40.9 | 26.2 | 15.7 | 8.9 |
| | BRONet (Lai et al., 2025) | 75M | 41.2 | 29.0 | 19.0 | 12.1 |
| | LipNeXt L32W1024 | 64M | 42.5 | 32.0 | 21.8 | 15.2 |
| | LipNeXt L32W2048 | 256M | **45.5** | **35.0** | **25.9** | **18.0** |

Table 2: Comparison of clean and certified accuracy using extra diffusion generated data from. Results of other methods are reported by Lai et al. (2025). The best results are marked in bold.

| Lipschitz Backbone | #Param. | CIFAR-10 | | | | CIFAR-100 | | | |
|---|---|---|---|---|---|---|---|---|---|
| | | Clean | $36/255$ | $72/255$ | $108/255$ | Clean | $\frac{36}{255}$ | $72/255$ | $108/255$ |
| LOT | 59M | 85.7 | 76.4 | 65.1 | 52.2 | 59.4 | 47.6 | 36.6 | 26.3 |
| Cayley | 68M | 86.7 | 77.7 | 66.9 | 54.3 | 61.1 | 48.7 | 37.8 | 27.5 |
| Cholesky | 68M | 85.4 | 76.6 | 65.7 | 53.3 | 59.4 | 47.4 | 36.8 | 26.9 |
| SLL | 83M | 85.6 | 76.8 | 66.0 | 53.3 | 59.4 | 47.6 | 36.6 | 27.0 |
| SOC | 83M | 86.6 | 78.2 | 67.0 | 54.1 | 60.9 | 48.9 | 37.6 | 27.8 |
| Lip-reg | 83M | 86.7 | 78.1 | 67.0 | 54.2 | 61.1 | 48.9 | 37.5 | 27.6 |
| BRO | 68M | 87.2 | 78.3 | 67.4 | 54.5 | 61.6 | 49.1 | 37.7 | 27.2 |
| Ours L32W2048 | 256M | 88.2 | 79.2 | 68.0 | 54.9 | 62.1 | 51.2 | 38.5 | 27.5 |
| Ours L32W2896 | 512M | **92.7** | **81.7** | **68.6** | **55.8** | **63.6** | **55.2** | **39.2** | **28.3** |

**Performance on ImageNet**  The main criticism of Lipschitz-based certification is the poor performance on large-scale datasets like ImageNet. We show that LipNeXt is feasible to scale up and achieve non-vanishing performance as model size increases. We consider two certification radius $\varepsilon = 36/255$, widely used for Lipschitz based method and $\varepsilon = 1$, widely used for randomized smoothing based method. Since the difference between the two radii is large, we train two models with different hyper-parameters. Specifically, the maximum training radius is set as 1.5 times of the test radius. Table 3 shows the comparison with prior work. By scaling to billon-level large models, LipNeXt outperforms prior work by 8% for $\varepsilon = 1$ and 3% for $\varepsilon = 36/255$.

Table 3: Comparison of clean accuracy and CRA on ImageNet. "Use EMD" indicates training with additional diffusion-generated data, and training speed is reported in minutes per epoch.

| Lipschitz Model | #Param. | Training Speed | Use EDM | $\varepsilon = 1$ | | $\varepsilon = {}^{36}/_{255}$ | |
|---|---|---|---|---|---|---|---|
| | | | | Clean Acc. | CRA | Clean Acc. | CRA |
| LiResNet | 51M | 5.3 | ✗ | 18.8 | 14.2 | 45.6 | 35.0 |
| LiResNet++ | 83M | - | ✓ | - | - | 49.0 | 38.3 |
| BRONet | 86M | 10.5 | ✗ | - | - | 49.3 | 37.6 |
| BRONet | 86M | - | ✓ | - | - | 52.3 | 40.7 |
| Ours L32W4096 | 1B | 8.9 | ✗ | 40.2 | 21.1 | 55.9 | 40.3 |
| Ours L32W5792 | 2B | 17.8 | ✗ | **41.0** | **22.4** | **57.0** | **41.2** |

Table (a): Fix depth as 32

| width | Acc. | CRA. |
|---|---|---|
| 1024 | 40.5 | 22.9 |
| 1456 | 43.5 | 24.5 |
| 2048 | 45.8 | 26.2 |
| 2896 | 48.5 | 28.2 |
| 4096 | 51.7 | 30.0 |

Table (b): Fix width as 2048

| depth | Acc. | CRA. |
|---|---|---|
| 8 | 30.7 | 22.4 |
| 16 | 43.5 | 24.7 |
| 32 | 45.8 | 26.2 |
| 64 | 47.0 | 26.9 |
| 128 | 47.5 | 26.8 |

Table (c): Fix #Param as 1B

| depth, width | Acc. | CRA. |
|---|---|---|
| (8, 8192) | 49.5 | 28.6 |
| (16, 5792) | 50.4 | 29.2 |
| (32, 4096) | 51.7 | 30.0 |
| (64, 2896) | 51.2 | 29.6 |
| (128, 2048) | 50.1 | 28.9 |

Table 4: Clean accuracy and CRA at $\varepsilon = 1$ on ImageNet (400 classes). Each sub-table fixes one factor (depth, width, or parameters) to study different configurations.

**Training stability and efficiency.** The wall-clock training time in Table 3 is measured using a single 8×H100 machine. The L32W5792 configuration is trained with two such machines, and we normalize the reported speed to an equivalent single-node throughput. LipNeXt adopts a constraint-free optimization whose forward and backward passes involve only orthogonal matrix multiplications and spatial shift operations, and we train it using `bfloat16`. By contrast, LiResNet employs power iteration to estimate per-block Lipschitz constants and regularizes the backbone Lipschitz (the product across blocks) in the loss. On ImageNet we observe numerical instability when training LiResNet with `bfloat16` (numeric overflow occurs occasionally), so we can only use `float32` for LiResNet instead. BRONet builds 1-Lipschitz convolutional networks via FFT-based frequency convolutions, which require complex arithmetic in both forward and backward passes. Because mainstream training frameworks currently support only `complex32`, this effectively incurs `float64`-like memory and compute overheads and prevents BRONet from fully leveraging low-precision accelerators. As a result, LipNeXt can continuously benefit from hardware advances due to its more stable architecture and optimization. Despite scaling to substantially larger models than LiResNet and BRONet, LipNeXt achieves training throughput on par with prior work, enabling further scaling. We leave training LipNeXt on large-scale image-text pair datasets as a future work.

**Scaling experiments** We demonstrate clear scaling trends for LipNeXt across both backbone depth and width dimensions. Due to computational constraints, we conduct experiments on a randomly sampled subset of 400 ImageNet classes, reporting clean accuracy and CRA at $\varepsilon = 1$ in Table 4. We systematically evaluate three scaling scenarios: (a) fixing backbone depth at 32 layers while varying width, (b) fixing backbone width at $W = 2048$ while varying depth, and (c) constraining the total parameter count to 1B while exploring different depth-width configurations. Our results reveal that LipNeXt exhibits favorable scaling properties with respect to both architectural dimensions, with performance improvements observed when increasing either depth or width. Notably, under fixed parameter budgets, a depth of 32 layers yields optimal performance.

**Ablation Studies** Due to space constraints, extended ablations are provided in Appendix C. Table 5 evaluates the two stabilization techniques for FastExp; Table 6 isolates the contribution of the spatial shift module; Table 7 compares padding and positional embedding choices; and Table 8 ablates activation functions.

# 5 CONCLUSION

We presented *LipNeXt*, a scalable 1-Lipschitz architecture that is both constraint-free and convolution-free. On the optimization side, we update orthogonal parameters directly on the manifold using a stabilized scheme, eliminating power-iteration penalties and avoiding the numerical fragility that inhibits low-precision training in prior work. On the architectural side, we replace depthwise convolutions with a theoretically grounded Spatial Shift Module. Together, these choices enable LipNeXt to scale cleanly in depth and width and to realize consistent gains in both clean accuracy and CRA. Empirically, LipNeXt establishes new state of the art across CIFAR-10/100, Tiny-ImageNet, and ImageNet with billion-parameter models, while maintaining competitive throughput and stable bfloat16 training. These results indicate that deterministic, Lipschitz-based certification can track modern scaling trends and further scale up.

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

# A  ADDITIONAL RELATED WORK

In contrast to the deterministic robustness guarantees emphasized in this work, *Randomized Smoothing* (RS) (Cohen et al., 2019a) provides probabilistic guarantees and has been extensively studied at ImageNet scale (Salman et al., 2019; Jeong et al., 2021; Salman et al., 2020). Diffusion methods (Carlini et al., 2022; Xiao et al., 2022) are introduced to denoise the addictive noise and further improve the performance.

Despite these successes, RS methods face two fundamental limitations that constrain their practical applicability. First, their inherently probabilistic nature introduces the possibility of false positives, where adversarial examples may be incorrectly certified as robust. While existing RS approaches typically maintain false positive rates below 0.1%, even this level of uncertainty renders them unsuitable for security-critical applications where certification guarantees must be absolute. Although one can reduce the false positive rates by improve the confidence level $\alpha$, it will then require significant greater number of noised images, which leads to the second limitation: the computational overhead of RS-based certification is prohibitive, necessitating thousands to tens of thousands of forward passes per image due to reliance on concentration inequalities with tight tail bounds. This computational burden has confined most empirical evaluations to small test subsets of at most 1,000 images, limiting the scope of practical validation.

# B  ADDITIONAL METHOD EXPLANATION

Algorithm 2 shows the naive Adam Optimization. The major issue of this optimization is the high computational cost of matrix exponential.

---

**Algorithm 2** Manifold Adam Optimizer

---

1: **Input:** learning rate $\eta$, momentum coefficients $\beta_1$ and $\beta_2$, optimization objective $f(X)$.
2: Initialize $X$ as an orthonormal matrix, and the first and second moment $m = v = 0$
3: **for** step $t$ in $1 \cdots, T$ **do**
4:   Compute the Euclidean gradient $\nabla f(X)$;
5:   Compute the projected gradient $\operatorname{grad} f(\mathbf{X})$ using $\nabla f(X)$ and $X$ by Equation 2;
6:   Update the first order moment: $m \leftarrow \beta_1 m + (1 - \beta_1)\operatorname{grad} f(\mathbf{X})$;
7:   Update the second order moment: $v \leftarrow \beta_2 v + (1 - \beta_2)\operatorname{grad} f(\mathbf{X}) * \operatorname{grad} f(\mathbf{X})$;
8:   Rescale $\hat{m} \leftarrow m/(1 - \beta_1^t)$ $\hat{v} \leftarrow v/(1 - \beta_2^t)$;
9:   Update the weights: $X \leftarrow X \exp(-\eta \cdot \hat{m}/(\sqrt{\hat{v}} + \epsilon)$
10: **end for**

---

## B.1  FURTHER DISCUSSION ABOUT $\beta$-ABS ACTIVATION

We show that the commonly used MinMax activation can be expressed by $\beta$-Abs if $\beta = 0.5$.

**Theorem 2** *Consider 2d-dimensional input $x = (x_1, x_2)^\top$ where $x_1, x_2 \in \mathbb{R}^d$. Define*

$$R = \frac{1}{\sqrt{2}} \begin{bmatrix} I_d & -I_d \\ I_d & I_d \end{bmatrix},$$

*Let $\beta = 0.5$, we have*

$$MinMax(x) = R^\top \beta\text{-}Abs(Rx)$$

**Proof:** First

$$\beta\text{-Abs}(Rx) = \beta\text{-Abs}(\frac{1}{\sqrt{2}} \begin{bmatrix} x_1 - x_2 \\ x_1 + x_2 \end{bmatrix}) = \frac{1}{\sqrt{2}} \begin{bmatrix} |x_1 - x_2| \\ x_1 + x_2 \end{bmatrix}$$

Then

$$R^\top \beta\text{-Abs}(Rx) = \frac{1}{2} \begin{bmatrix} |x_1 - x_2| + x_1 + x_2 \\ -|x_1 - x_2| + x_1 + x_2 \end{bmatrix} = \begin{bmatrix} \max(x_1, x_2) \\ \min(x_1, x_2) \end{bmatrix}.$$

Here $\max$ and $\min$ are element wise operations for vectors. Theorem 2 shows that **MinMax** and $\beta$-Abs should have the same expressive ability. Increasing $\beta$ introduces more non-linearity and decreasing $\beta$ introduces more linearity. However, **MinMax** corresponds to a fixed non-linearity, whereas $\beta$-Abs allows continuous interpolation between linear and piecewise-linear regimes.

## B.2 PyTorch-like Code for the LipNeXt block

Below is a PyTorch-like Code for the LipNeXt block for a better understanding.

```
def shift_fn(x, alpha=1/16):
    c = x.shape[3]
    d = int(c * alpha)
    a0, a1, a2, a3, a4 = torch.split(
        x, [c - d * 4, d, d, d, d], dim=3)

    a1 = torch.roll(a1, dims=1, shifts=1)
    a2 = torch.roll(a2, dims=1, shifts=-1)
    a3 = torch.roll(a3, dims=2, shifts=1)
    a4 = torch.roll(a4, dims=2, shifts=-1)

    x = torch.cat([a0, a1, a2, a3, a4], dim=3)
    return x

def beta_abs(x, beta=0.75):
    d = int(x.shape[1] * beta)
    x = torch.cat([x[..., :d].abs(), x[..., d:]], dim=-1)
    return x

def lipnext_block(x, R, M, b, pos)
    # shape of x: (B, H, W, C)
    # shape of R and M: (C, C)
    # shape of b: (C)
    # shape of pos: (H, W, 1)
    x = x + pos
    x = F.linear(x, R)
    x = shift_fn(x)
    x = F.linear(x, W @ R.T, b)
    x = beta_abs(x)
    return x
```

## B.3 Proof of Theorem 1

**Part 1: ($\Leftarrow$) Sufficiency**

Assume the kernel $K$ contains exactly one non-zero element with value $c = \pm 1$. Let this element be at position $(i_0, j_0)$, so $K_{i_0,j_0} = c$ and all other elements of $K$ are zero.

The convolution $f_K$ is a linear operator. Therefore, we can write:

$$\|f_K(X) - f_K(Y)\|_F = \|f_K(X - Y)\|_F$$

Let $Z = X - Y$. We need to show that $\|f_K(Z)\|_F = \|Z\|_F$.

The result of the convolution, let's call it $W = f_K(Z)$, is given by:

$$W_{i,j} = \sum_{m=0}^{k-1} \sum_{n=0}^{k-1} K_{m,n} Z_{i-m,j-n}$$

where the indices for $Z$ are taken modulo $H$ and $W$ due to circular padding. Since only $K_{i_0,j_0}$ is non-zero, this sum simplifies to a single term:

$$W_{i,j} = K_{i_0,j_0} Z_{i-i_0,j-j_0} = c \cdot Z_{i-i_0,j-j_0}$$

This means the output tensor $W$ is a circularly shifted version of the input tensor $Z$, with each element multiplied by $c$.

Now, let's compute the squared Frobenius norm of $W$:

$$\|W\|_F^2 = \sum_{i=0}^{H-1} \sum_{j=0}^{W-1} |W_{i,j}|^2 = \sum_{i=0}^{H-1} \sum_{j=0}^{W-1} |c \cdot Z_{i-i_0,j-j_0}|^2$$

Since $c = \pm 1$, we have $c^2 = 1$.

$$\|W\|_F^2 = \sum_{i=0}^{H-1} \sum_{j=0}^{W-1} Z_{i-i_0, j-j_0}^2$$

The mapping $(i, j) \mapsto (i - i_0 \pmod{H}, j - j_0 \pmod{W})$ is a bijection on the set of indices. Therefore, the sum on the right is simply a reordering of the sum of the squared elements of $Z$.

$$\sum_{i=0}^{H-1} \sum_{j=0}^{W-1} Z_{i-i_0, j-j_0}^2 = \sum_{i'=0}^{H-1} \sum_{j'=0}^{W-1} Z_{i', j'}^2 = \|Z\|_F^2$$

Thus, we have shown that $\|f_K(Z)\|_F^2 = \|Z\|_F^2$, which implies $\|f_K(Z)\|_F = \|Z\|_F$. This completes the first part of the proof.

**Part 2: ($\Rightarrow$) Necessity**

Assume that $f_K$ is norm-preserving, i.e., $\|f_K(X) - f_K(Y)\|_F = \|X - Y\|_F$ for all $X, Y \in \mathbb{R}^{H \times W}$. As before, let $Z = X - Y$. The condition is equivalent to stating that $f_K$ is a linear isometry with respect to the Frobenius norm:

$$\|f_K(Z)\|_F = \|Z\|_F, \quad \forall Z \in \mathbb{R}^{H \times W}$$

We analyze this condition in the frequency domain. Let $\mathcal{F}$ denote the 2D Discrete Fourier Transform (DFT), and let $\hat{A} = \mathcal{F}(A)$. The convolution theorem for circular convolution states:

$$\mathcal{F}(f_K(Z)) = \hat{K}' \odot \hat{Z}$$

where $K'$ is the kernel $K$ zero-padded to size $H \times W$, and $\odot$ denotes element-wise (Hadamard) product.

Parseval's theorem relates the Frobenius norm of a tensor to the Frobenius norm of its DFT:

$$\|A\|_F^2 = \frac{1}{HW} \|\hat{A}\|_F^2$$

Applying Parseval's theorem to our isometry condition $\|f_K(Z)\|_F^2 = \|Z\|_F^2$:

$$\frac{1}{HW} \|\mathcal{F}(f_K(Z))\|_F^2 = \frac{1}{HW} \|\hat{Z}\|_F^2$$

$$\|\hat{K}' \odot \hat{Z}\|_F^2 = \|\hat{Z}\|_F^2$$

Expanding the norms in terms of their elements:

$$\sum_{u=0}^{H-1} \sum_{v=0}^{W-1} |\hat{K}'_{u,v} \cdot \hat{Z}_{u,v}|^2 = \sum_{u=0}^{H-1} \sum_{v=0}^{W-1} |\hat{Z}_{u,v}|^2$$

$$\sum_{u=0}^{H-1} \sum_{v=0}^{W-1} |\hat{K}'_{u,v}|^2 |\hat{Z}_{u,v}|^2 = \sum_{u=0}^{H-1} \sum_{v=0}^{W-1} |\hat{Z}_{u,v}|^2$$

This can be rewritten as:

$$\sum_{u=0}^{H-1} \sum_{v=0}^{W-1} \left( |\hat{K}'_{u,v}|^2 - 1 \right) |\hat{Z}_{u,v}|^2 = 0$$

Since this equality must hold for any tensor $Z$, it must hold for any possible DFT $\hat{Z}$. Let us choose a $\hat{Z}$ that has only one non-zero element, say $\hat{Z}_{u_0, v_0} = 1$ and all other elements are zero. For this choice, the equation simplifies to:

$$\left( |\hat{K}'_{u_0, v_0}|^2 - 1 \right) \cdot 1^2 = 0 \implies |\hat{K}'_{u_0, v_0}|^2 = 1$$

As we can make this choice for any frequency pair $(u_0, v_0)$, it must be that $|\hat{K}'_{u,v}| = 1$ for all $u, v$.

Now we use this property to derive constraints on the spatial domain kernel $K$.

1. **Sum of Squares of Elements**: Apply Parseval's theorem to the padded kernel $K'$ itself:

$$\|K\|_F^2 = \|K'\|_F^2 = \frac{1}{HW}\|\hat{K}'\|_F^2 = \frac{1}{HW}\sum_{u=0}^{H-1}\sum_{v=0}^{W-1}|\hat{K}'_{u,v}|^2$$

Since we found that $|\hat{K}'_{u,v}|^2 = 1$ for all $(u,v)$, the sum becomes $\sum_{u,v} 1 = HW$.

$$\|K\|_F^2 = \frac{1}{HW}(HW) = 1 \implies \sum_{i,j} K_{i,j}^2 = 1$$

2. **Sum of Elements**: Consider the DC component of the DFT, i.e., $(u,v) = (0,0)$:

$$\hat{K}'_{0,0} = \sum_{i=0}^{H-1}\sum_{j=0}^{W-1} K'_{i,j}e^{-0} = \sum_{i=0}^{k-1}\sum_{j=0}^{k-1} K_{i,j}$$

Since $|\hat{K}'_{0,0}| = 1$, we must have:

$$\left|\sum_{i,j} K_{i,j}\right| = 1$$

Let $\{k_1, k_2, \ldots, k_N\}$ be the set of $N$ non-zero elements in the kernel $K$. From our derivations, we have two conditions on these elements:

1. $\sum_{i=1}^{N} k_i^2 = 1$
2. $\left(\sum_{i=1}^{N} k_i\right)^2 = 1$

We can expand the second condition:

$$\left(\sum_{i=1}^{N} k_i\right)^2 = \sum_{i=1}^{N} k_i^2 + 2\sum_{1 \leq i < j \leq N} k_i k_j = 1$$

Substituting the first condition into this expansion:

$$1 + 2\sum_{1 \leq i < j \leq N} k_i k_j = 1 \implies \sum_{1 \leq i < j \leq N} k_i k_j = 0$$

We now have $\sum k_i^2 = 1$ and $\sum k_i^2 = (\sum k_i)^2$. Let's consider the Cauchy-Schwarz inequality on the vectors $\mathbf{a} = (1, 1, \ldots, 1) \in \mathbb{R}^N$ and $\mathbf{b} = (k_1, k_2, \ldots, k_N) \in \mathbb{R}^N$:

$$(\mathbf{a} \cdot \mathbf{b})^2 \leq \|\mathbf{a}\|_2^2 \|\mathbf{b}\|_2^2$$

$$\left(\sum_{i=1}^{N} k_i\right)^2 \leq \left(\sum_{i=1}^{N} 1^2\right)\left(\sum_{i=1}^{N} k_i^2\right)$$

Plugging in our conditions:

$$1 \leq N \cdot 1$$

Equality holds if and only if one vector is a scalar multiple of the other, i.e., $\mathbf{b} = c\mathbf{a}$ for some scalar $c$. This means all non-zero elements must be equal: $k_1 = k_2 = \cdots = k_N = c$.

Let's impose this equality on our conditions:

1. $\sum_{i=1}^{N} c^2 = Nc^2 = 1$
2. $\left(\sum_{i=1}^{N} c\right)^2 = (Nc)^2 = N^2 c^2 = 1$

Substituting $c^2 = 1/N$ from the first equation into the second gives:

$$N^2\left(\frac{1}{N}\right) = N = 1$$

This shows that there must be exactly one non-zero element ($N = 1$).

Let this single non-zero element be $k_1$. From condition 1, $k_1^2 = 1$, which implies $k_1 = \pm 1$. Therefore, for $f_K$ to be norm-preserving, the kernel $K$ must contain exactly one non-zero element with a value of $+1$ or $-1$. This completes the second part of the proof.

## C    ADDITIONAL RESULTS

Table 5: Ablation study on the effectiveness of the two stabilization techniques for FastExp.

| LipNeXt L32W2048 | CIFAR-10 | | | | CIFAR-100 | | | |
|---|---|---|---|---|---|---|---|---|
| | Clean | $36/255$ | $72/255$ | $108/255$ | Clean | $\frac{36}{255}$ | $72/255$ | $108/255$ |
| Algorithm 1 | 85.0 | 73.2 | 58.8 | 43.3 | 57.4 | 44.1 | 31.9 | 22.2 |
| No Periodic polar retraction | 84.8 | 72.3 | 57.2 | 42.0 | 57.2 | 43.3 | 31.4 | 21.3 |
| No Lookahead | 84.4 | 72.4 | 57.8 | 42.7 | 57.4 | 43.5 | 31.4 | 21.7 |

Table 5 shows the effectiveness of the two stabilization techniques for FastExp. Removing either would lead to constant performance drop.

Table 6: Ablation study on the effectiveness of the spatial shift module

| LipNeXt L32W2048 | CIFAR-10 | | | | CIFAR-100 | | | |
|---|---|---|---|---|---|---|---|---|
| | Clean | $36/255$ | $72/255$ | $108/255$ | Clean | $\frac{36}{255}$ | $72/255$ | $108/255$ |
| $\alpha = 1/16$ (baseline) | 85.0 | 73.2 | 58.8 | 43.3 | 57.4 | 44.1 | 31.9 | 22.2 |
| $\alpha = 0$ (no shift) | 62.2 | 53.6 | 43.7 | 32.2 | 40.8 | 32.0 | 22.3 | 14.3 |
| $\alpha = 1/4$ (shift all channels) | 79.0 | 67.4 | 51.4 | 38.3 | 52.5 | 39.1 | 26.4 | 18.2 |

Table 6 shows the effectiveness of the spatial shift module. Setting $\alpha = 1/16$ shifts one quarter of the channels (distributed across four directions). We examine two extremes: $\alpha = 0$, which applies no spatial shift, and $\alpha = 1/4$, which shifts all channels. As expected, $\alpha = 0$ performs poorly because the model cannot capture spatial interactions. Interestingly, $\alpha = 1/4$ also degrades performance; we hypothesize that shifting every channel removes absolute positional cues, preventing the model from retaining information about the original locations.

Table 7: Ablation study on the design choices of the padding and positional embedding

| Padding Type | Positional Encoding | CIFAR10 clean Acc. | CIFAR10 CRA | CIFAR100 clean Acc. | CIFAR100 CRA |
|---|---|---|---|---|---|
| Circular | ✓ | 85.0 | 73.2 | 57.4 | 44.1 |
| Circular | ✗ | 84.3 | 72.1 | 56.2 | 43.2 |
| Zero | ✗ | 84.5 | 72.4 | 56.5 | 43.5 |
| Zero | ✓ | 84.6 | 72.6 | 56.8 | 43.7 |

Table 7 presents an ablation of padding and positional encoding choices. As discussed in Section 3, without positional encoding, zero padding outperforms circular padding because it implicitly introduces positional information. With positional encoding enabled, the trend reverses: circular padding is preferable since it guarantees a Lipschitz-tight transformation.

Table 8: Ablation study on the choice of activations

| LipNeXt L32W2048 | CIFAR-10 | | | | CIFAR-100 | | | |
|---|---|---|---|---|---|---|---|---|
| | Clean | $36/255$ | $72/255$ | $108/255$ | Clean | $\frac{36}{255}$ | $72/255$ | $108/255$ |
| MinMax | 84.4 | 72.7 | 58.0 | 42.6 | 56.9 | 43.5 | 31.2 | 21.4 |
| $\beta = 0.5$ | 84.6 | 72.9 | 58.3 | 42.9 | 57.1 | 43.8 | 31.5 | 21.7 |
| $\beta = 0.75$ (default) | 85.0 | 73.2 | 58.8 | 43.3 | 57.4 | 44.1 | 31.9 | 22.2 |
| $\beta = 1.0$ | 84.3 | 72.5 | 58.1 | 42.3 | 56.5 | 43.2 | 31.0 | 20.8 |

Table 8 ablates the activations. As we shown in Appendix B.1, $\beta$-Abs and MinMax should have the same expressive ability when $\beta = 0.5$. Experiments show that MinMax is slighter worse than $\beta = 0.5$. We can adjust $\beta$ to control the non-linearity, and $\beta = 1$ degrades to the absolution function is not optimal because it breaks identical mapping.

