# OpenReview forum: "LipNeXt: Scaling up Lipschitz-based Certified Robustness to Billion-parameter Models"
_ICLR.cc/2026/Conference — ICLR 2026 Poster_

### Official Review · Reviewer_ZYaL · 2025-10-16

**Soundness:** 3
**Presentation:** 3
**Contribution:** 3
**Rating:** 8
**Confidence:** 4

**Summary:**

This paper studies certified robustness in the direction of Lipschitz networks. Algorithmically, it (1) designs an approximate-then-correct optimization algorithm which speeds up the optimization in the orthogonal space, and (2) proposes a convolution-free method via shifting (rolling) data spatially in the orthogonal embedding space. The final method is able to scale to networks with billion parameters and achieves strong performance.

**Strengths:**

The proposed method exhibits strong rigor and novelty. Almost all designs are supported by strong theoretical analysis and well-motivated. The final overall performance demonstrates the effectiveness of the general algorithm. Detailed ablation studies are provided in the appendix to demonstrate the effectiveness of individual modules.

**Weaknesses:**

The overall algorithm seems costly. On the memory side, the optimizer requires a copy of the full parameter, thus doubles the memory cost, which is especially concerning for a model with billion-parameters. On the computation side, the main results on conducted on 8xH100 GPUs, which seems hard to reproduce by academic labs and not scalable to harder tasks. I will not attack the main contribution due to the costs though.

The parameter efficiency is of question. All comparisons, although meaningful in the story, are conducted with regard to models with far fewer parameters. I will not deny that it is meaningful to further scale and the baselines might suffer from scaling, but the key question is, are those parameters necessary to achieve strong performance; which part of the model consumes the most parameters?

As a minor comment, the introduction about certified training is misleading. Line 43 introduces randomized smoothing and Lipshitz networks as the two major trends, while other deterministic certified training based on bound propagation is as active, see [1] as an example.

[1] https://arxiv.org/abs/2406.04848

**Questions:**

1. An orthogonal projection, $R$, is conducted before the spatial shifting. Is this a learned orthogonal matrix or a manually constructed matrix?

2. Is the spatial shifting method meaningful without projecting into an orthogonal space first? The current design arguably introduces much more parameters (the $R$ matrix is a full-dimensional square matrix) than convolutions.

---

> ### Author Response · Authors · 2025-11-25
> **Response to Reviewer ZYaL**
>
> Thank you for your insightful comments and thorough review. We address your concerns as follows:
> ## Weaknesses:
> >The optimizer requires a copy of the full parameter
>
> This memory overhead can be effectively mitigated using the Zero Redundancy Optimizer (ZeRO) [1], which partitions the optimizer states across GPUs rather than replicating them. For instance, when training on an 8-GPU machine with 32 large parameter matrices, each GPU only stores 4 matrices rather than all 32. This approach is widely used to train large language models and significantly reduces the per-GPU memory while maintaining training efficiency
>
> [1] Rajbhandari et al: ZeRO: Memory Optimizations Toward Training Trillion Parameter Models
>
> >The main results on conducted on 8xH100 GPUs
>
> We would like to clarify that our results are fully reproducible on more modest hardware configurations. The table below demonstrates the wall-clock time and CUDA memory when running our method on CIFAR10 using a 2×A100 machine:
> | LipNeXt config | batch size per GPU | wall clock time (mins/epoch) | CUDA memory (GB) |
> |:-:|:-:|:-:|:--:|
> |L32W1024 | 256 | 0.47  | 16.4 |
> |L32W2048 | 256 | 1.33 | 35.6 |
> |L32W2896 | 256 | 2.43 | 56.8 |
>
> The CUDA memory can be further reduced by using a smaller batch size and gradient accumulation. As our method does not orthogonalize the weight matrix during model forward, applying gradient accumulation does not incur additional overhead
>
> >Comparisons are conducted with models with far fewer parameters
>
> We respectfully note that Table 1 in the main paper presents comparisons at matched model sizes, where our method demonstrates superior performance. We have also attempted to scale competing methods to larger sizes; however, they exhibit scalability limitations as shown in the table below. The performance metric is ``clean accuracy`` and ``CRA`` at $\varepsilon=36/255$
>
> | Model    | Config     | Size (M) | CIFAR-10    | CIFAR-100   |
> |:-:|:-:|:-:|:-:|:-:|
> | LipNeXt  | L32W2048   | 256 | 88.2/79.2   | 62.1/51.2   |
> | LipNeXt  | L32W2896   | 512 | 92.7/81.7   | 63.6/55.2   |
> | LiResNet | L12W1250   | 256 | 87.3/78.4   | 61.4/49.4   |
> | LiResNet | L12W1888   | 512 | 87.2/78.4   | 61.2/49.5   |
> | BRONet   | L12W1300   | 256 | OOM         | OOM         |
>
> As shown, LiResNet exhibits marginal improvement when scaled, while BRONet encounters out-of-memory errors at the 256M parameter scale, even at batch size 1 per GPU. These results highlight the superior scalability of our approach
>
> >Are those parameters necessary to achieve strong performance
>
> This is an excellent and fundamental question. While a definitive answer requires discovering more parameter-efficient solutions, we offer the following perspective:
>
> Certified robustness fundamentally requires more computational capacity than standard prediction. Specifically, certified predictions provide guarantees for all inputs within an $\varepsilon$-ball around x (i.e., predictions for all x' where $\|x-x'\|\leq\varepsilon$), which is significantly more information than a single point prediction. Different approaches distribute this computational cost differently:
> - Randomized Smoothing: Amortizes cost using test time scaling
> - Lipschitz-based methods: Amortizes cost using parameter scaling
> Both approaches require substantial resources and our method shifts the burden from inference to training
>
> >Which part of the model consumes the most parameters
>
> The primary parameter consumption comes from the large projection matrices R and M in Equation 9
>
> > Line 43 introduces randomized smoothing and Lipschitz networks as the two major trends, while other deterministic certified training based on bound propagation is as active
>
> Thank you for highlighting this omission. We agree that bound propagation methods are indeed widely used, particularly for $\ell_\infty$ certification. We will incorporate a discussion of this important line of work in our related work
>
> ## Questions
> > An orthogonal projection is conducted before the spatial shifting. Is this a learned orthogonal matrix or manually constructed
>
> R is a learned orthogonal matrix parameter (along with M in Equation 9). We have also experimented with fixed (non-learned) orthogonal matrices initialized randomly, which reduces model size by approximately 50% (e.g., from 64M to 32M parameters). However, our experiments show that the learned version consistently outperforms the fixed version when controlling for parameter count
>
> >Is the spatial shifting method meaningful without projecting into an orthogonal space first
>
> The orthogonal projection is essential for the effectiveness of spatial shifting. Without the R projection, the spatial shifting operation would consistently shift the same subset of channels, severely limiting the representational capacity. The projection R performs a learned channel permutation/mixing, while R^T reverses this transformation. This mechanism ensures that different feature channels are shifted across different layers

---

> > ### Comment · Reviewer_ZYaL · 2025-11-26
> >
> > Thanks for the detailed reply. My concerns are all clear.

---

### Official Review · Reviewer_cWBh · 2025-10-30

**Soundness:** 3
**Presentation:** 3
**Contribution:** 3
**Rating:** 6
**Confidence:** 2

**Summary:**

LipNeXt introduces a novel 1-Lipschitz architecture that provides efficient, deterministic robustness guarantees for large models. By leveraging manifold optimization and a Spatial Shift Module, it achieves state-of-the-art certified robustness accuracy on CIFAR-10, CIFAR-100, Tiny-ImageNet, and ImageNet.

**Strengths:**

The paper provides solid empirical evidence, along with an ablation study, to support the main claims.

**Weaknesses:**

Table 2 presents results with additional data; however, I noticed that the total number of parameters for the proposed model is 256M, which is significantly larger than the competitors. Could the authors provide results for a smaller model configuration, such as L32W1024?

**Questions:**

* Could the authors provide a formal definition of certified robust accuracy (CRA) and clarify how the corresponding data is generated? As mentioned in line 40, ε represents the radius of a ball in the p-norm. However, the specific value of p does not seem to be provided. Additionally, since the introduction begins by discussing adversarial robustness, I assume that CRA refers to accuracy against adversarial examples. Could the authors clarify which attack methods were used for the assessments?

* I find the role of the β-Abs nonlinearity somewhat unclear. It is introduced between lines 320-357 as a replacement for MinMax or MaxMin. While I do not doubt the correctness of the proof, the connection to the overall proposed method is not immediately clear from a narrative perspective. Additionally, the improvement gains from β-Abs, as shown in Table 7, appear to be relatively modest compared to those in Table 1. Besides, could the authors clarify whether β-Abs could be applied to existing models, and if so, under what conditions?

---

> ### Author Response · Authors · 2025-11-26
> **Response to Reviewer cWBh (1/2)**
>
> Thank you for your insightful comments and thorough review. We address your concerns as follows:
>
> ## Weaknesses:
> > Could the authors provide results for a smaller model configuration, such as L32W1024?
>
> Thank you for this suggestion. We have conducted additional experiments with a smaller configuration (L32W1024) as well as larger configurations for baseline methods. The results are summarized in the table below, where we report clean accuracy and CRA at $\varepsilon=36/255$:
> | Model| Config| Size (M) | CIFAR-10| CIFAR-100|
> |:-:|:-:|:-:|:-:|:-:|
> | LipNeXt |L32W1024|64|85.6/76.2|59.3/47.3|
> | LipNeXt|L32W2048|256|88.2/79.2|62.1/51.2|
> | LipNeXt|L32W2896|512|92.7/81.7|63.6/55.2|
> ||
> |LiResNet|L12W512| 83 | 86.7/78.4| 61.1/49.2|
> |LiResNet|L12W1250| 256 | 87.3/78.4| 61.4/49.4|
> |LiResNet|L12W1888| 512 | 87.2/78.4| 61.2/49.5|
> ||
> |BRONet|L12W512|64| 87.2/78.3|61.6/49.1|
> |BRONet|L12W1300|256|OOM|OOM|
>
> Analysis: The smaller L32W1024 LipNeXt variant shows modest performance gains when trained on diffusion-generated data. This is primarily due to our use of the spatial shift module for computational efficiency, which trades some model capacity compared to convolution-based methods with equivalent parameter counts. However, LipNeXt demonstrates superior scalability compared to baselines when increasing model size. This advantage stems from two key factors:
> - A tighter Lipschitz bound achieved through the use of orthogonal operators throughout the architecture
> - An improved loss landscape resulting from direct optimization on the orthogonal manifold
>
> Notably, LiResNet exhibits only marginal improvement when scaled, while BRONet encounters out-of-memory errors at the 256M parameter scale, even with batch size 1 per GPU. This limitation arises because BRONet uses FFT to construct orthogonal convolutions, which requires O(BHWC²) CUDA memory for input size B×C×H×W.
>
> ## Questions 1:
> > Could the authors provide a formal definition of certified robust accuracy (CRA)?
>
> CRA is defined as the percentage of test examples for which the prediction is correct and the model can provide a rigorous mathematical proof demonstrating that the prediction is invariant to all perturbations within an $\ell_p$ ball.
>
> The mathematical proof varies across different methods (e.g., randomized smoothing vs. Lipschitz-based methods). Our method follows the Lipschitz-based approach, as detailed in Section 2.1. Specifically, given:
> - $K\in\mathbb{R}$ denote an Lipschitz estimation of the model backbone,
> - $W\in\mathbb{R}^{n\times d}$ denote the n-class classification head,
> - $W[i]$ denote the $i$-th row of $W$,
> - $y\in\mathbb{R}^{n}$ denote the output logit of the model for a given input $x$,
> - $T$ denote the ground truth label.
> A prediction is certified robust if $\forall 1\leq j\leq n, \quad y_T > y_j + \|W_T - W_j\|_2 K \varepsilon$
>
> > The specific value of p does not seem to be provided
>
> We apologize for this omission. This work focuses on p=2 (ℓ₂ norm). We will clarify this in the revision
>
> > I assume that CRA refers to accuracy against adversarial examples. Could the authors clarify which attack methods were used for the assessments?
>
> We appreciate the opportunity to clarify this important distinction. CRA computation does not involve any attack method—this is precisely the advantage of certified robustness over empirical robustness.
>
> Key distinction: Empirical adversarial robustness depends on the specific attack method used for evaluation. If a stronger attack is proposed in the future, a prediction deemed empirically robust may no longer hold. In contrast, certified robustness provides a mathematical guarantee: once a prediction is proven robust within a given perturbation budget, it remains robust regardless of any future attack methods.

---

> ### Author Response · Authors · 2025-11-26
> **Response to Reviewer cWBh (2/2)**
>
> ## Question 2
> > The connection to the overall proposed method is not immediately clear from a narrative perspective.
>
> We appreciate this feedback. The motivation for β-Abs is to provide a more GPU-efficient activation function while maintaining orthogonality. Prior activation functions have significant computational overhead:
>
> - MinMax: requires swapping values between tensor partitions
> - HouseHolder: requires 4 logical operations and 12 sin/cos operations
>
> In contrast, β-Abs uses only an in-place operation, resulting in substantial efficiency gains. The table below compares training speed (minutes/epoch) for LipNeXt on a 4×A100 machine:
> |Model Config |$\beta$-Abs | MinMax |HouseHolder |
> |:-:|:-:|:-:|:-:|
> |LipNeXt L32W1024| 0.18 | 0.23 | 0.79 |
> |LipNeXt L32W2048| 0.57 | 0.66 | 1.69 |
>
> β-Abs achieves approximately 10% training speedup compared to MinMax, without requiring any other architectural changes. We believe such GPU-efficiency property is preferred for model scaling up.
>
> > Could the authors clarify whether β-Abs could be applied to existing models, and if so, under what conditions?
>
> Yes, β-Abs can be readily applied to existing Lipschitz-constrained models as a drop-in replacement for the MinMax activation. As shown in Theorem 2, β-Abs is mathematically equivalent to MinMax with rotations when β=0.5, suggesting similar performance. Additionally, β-Abs provides greater flexibility to control the degree of non-linearity through the β parameter, and we find β=0.75 is even better.
>
> > the improvement gains from β-Abs, as shown in Table 7, appear to be relatively modest compared to those in Table 1.
>
> This observation is expected and aligns with our design philosophy. The primary contribution of our architecture is the spatial shift module, which enables GPU-efficient Lipschitz training. β-Abs serves as a complementary component that improves computational efficiency while maintaining comparable performance. Our overall performance gains (Table 1) result from the combination of model scaling, and enhanced training techniques, rather than from the architectural design, which contributes to efficiency.

---

### Official Review · Reviewer_BPtS · 2025-11-01

**Soundness:** 3
**Presentation:** 3
**Contribution:** 3
**Rating:** 6
**Confidence:** 4

**Summary:**

The paper proposes a new approach for constructing Lipschitz neural networks and demonstrates improved certified robustness compared with several baselines. The method primarily focuses on two key components. They are Manifold Optimization and Spatial Shift Module. The latter one is novel to me.
Overall, I lean toward accepting this paper. I may further increase my score if the authors can address my concerns properly.

**Strengths:**

1.	The paper is well-organized and clearly written.
2.	The proposed manifold optimization and spatial shift techniques are interesting and technically sound.

**Weaknesses:**

1.	Some important baselines are not discussed in the related work section, such as Sandwich and BRONet.
2.	The method integrates several known techniques, making it somewhat difficult to assess the individual effectiveness of each component.

**Questions:**

1.	I suggest separating Sections 3.1 and 3.2–3.3. This would allow the paper to present Sections 3.2–3.3 together with other orthogonal methods such as LOT, BRO, or Cholesky for comparison. I also recommend adding an ablation study combining those orthogonal baselines with Sections 3.2–3.3 to better highlight their contribution.
2.	In Table 2, the proposed model appears larger than the baseline. Could the authors include LipNeXt L32W1024 in the table for a fairer comparison? Additionally, I am curious about the scaling ability of the baselines, as the proposed method shows clear scaling behavior when synthetic data are used.
3.	In Table 3, could the authors provide the EDM results for the proposed method? Including these results would help illustrate how scaling up combined with EDM could further enhance performance.
4.	I do not fully understand the relationship between β-Abs and min–max. In Theorem 2, is it necessary to use this $R$? Also, please verify whether line 751 is correct.
5.	SOC introduces other activation functions such as Householder and GroupSort. Can the paper compare with these? Otherwise, the advantage of β-Abs is unclear.
6.	What does the dagger ($\dagger$ in line 351) symbol in Table 1 represent?
7.	Why does the paper use R in Equation (8)? Is this orthogonal mapping essential and trainable?
8.	The periodic polar retraction seems somewhat unusual. The paper claims it helps eliminate errors, but could the authors clarify its specific effect? Why not use other orthogonal projections such as LOT or Cholesky?
9.	It would strengthen the paper if the authors could include AutoAttack results to verify that the certified robustness implementation is correct.
10.	Regarding Section 3.1, I believe the following reference may be missing:
Bader, Philipp, Sergio Blanes, and Fernando Casas. "Computing the matrix exponential with an optimized Taylor polynomial approximation." Mathematics 7.12 (2019): 1174.

---

> ### Author Response · Authors · 2025-11-26
> **Response to Reviewer BPtS (1/2)**
>
> Thank you for your insightful comments and thorough review. We address your concerns as follows:
>
> ## Weaknesses
>
> >Some important baselines are not discussed in the related work section, such as Sandwich and BRONet.
>
> Thank you for highlighting this omission. We will incorporate a comprehensive discussion of these important baselines, including Sandwich and BRONet, in the related work section of the revised manuscript.
>
> >The method integrates several known techniques, making it somewhat difficult to assess the individual effectiveness of each component.
>
> We appreciate this concern. We have included ablation studies in Appendix C that examine the individual contributions of key components. If you would like us to evaluate any specific module not currently covered in Appendix C, we would be happy to provide additional ablation studies
>
> ## Question 1
>
> >separating Sections 3.1 and 3.2–3.3 with Sections 3.2–3.3
>
> Thank you for this suggestion. We agree that separating these sections would improve the clarity and flow of the methodology presentation. We will implement this restructuring in the final version
>
> >adding an ablation study combining those orthogonal baselines
>
> We have conducted an ablation study comparing different orthogonalization methods within our spatial shift architecture. The table below summarizes the results for L32W2048, reporting clean accuracy and Certified Robust Accuracy (CRA) at ε=36/255. Training speed is measured in minutes/epoch on a 4×A100 machine:
>
> | orthogonalization| CIFAR10 | CIFAR100 | training speed |
> |:-:|:-:|:-:|:-:|
> |Manifold optimization| 85.0/73.2 | 57.4/44.1 | 0.57 |
> |Cholesky Reparameterization | 83.2/71.0 | 56.9/42.8| 0.76|
> |Cayley Reparameterization | 82.4/70.8 | 56.2/42.6 | 0.88 |
> | BRO Reparameterization | 79.4/68.3| 52.3/38.9| 0.59 |
>
> **Note on BRO comparison:** A direct comparison with BRO is nuanced, as it utilizes only one quarter the number of parameters:
> $$y = (I - 2 V(V^\top V^{-1})V)x \text{ where } x\in \mathbb{R}^n, \quad V\in \mathbb{R}^{n \times n/4}$$
> The number of parameters is $\frac{1}{4}n^2$ while the computational cost is $\frac{1}{2}n^2$.
>
> Scaling BRO to match the parameter count would correspondingly double the computational cost. In this table, we control the computational cost because it is more important when scaling up the models
>
> ## Question 2
> >Could the authors include LipNeXt L32W1024 in the table for a fairer comparison? Additionally, I am curious about the scaling ability of the baselines
>
> Thank you for this valuable suggestion. We have conducted additional experiments with both a smaller configuration (L32W1024) and larger configurations for baseline methods. The results are summarized below, reporting clean accuracy and CRA at ε=36/255:
>
> | Model| Config| Size (M) | CIFAR-10| CIFAR-100|
> |:-:|:-:|:-:|:-:|:-:|
> | LipNeXt |L32W1024|64|85.6/76.2|59.3/47.3|
> | LipNeXt|L32W2048|256|88.2/79.2|62.1/51.2|
> | LipNeXt|L32W2896|512|92.7/81.7|63.6/55.2|
> ||
> |LiResNet|L12W512|83| 86.7/78.4|61.1/49.2|
> |LiResNet|L12W1250|256|87.3/78.4|61.4/49.4|
> |LiResNet|L12W1888|512|87.2/78.4|61.2/49.5|
> ||
> |BRONet|L12W512|64|87.2/78.3|61.6/49.1|
> |BRONet|L12W1300|256|OOM||
>
> Analysis: The smaller L32W1024 LipNeXt variant shows modest performance gains when trained on diffusion-generated data. This is primarily due to our use of the spatial shift module for computational efficiency, which trades some model capacity compared to convolution-based methods with equivalent parameter counts. However, LipNeXt demonstrates superior scalability compared to baselines when increasing model size. This advantage stems from two key factors:
> - A tighter Lipschitz bound achieved through the use of orthogonal operators throughout the architecture
> - An improved loss landscape resulting from direct optimization on the orthogonal manifold
>
> Notably, LiResNet exhibits only marginal improvement when scaled, while BRONet encounters out-of-memory errors at the 256M parameter scale, even with batch size 1 per GPU. This limitation arises because BRONet uses FFT to construct orthogonal convolutions, which requires O(BHWC²) CUDA memory for input size B×C×H×W
>
> ## Question 3
> >provide the EDM results for the proposed method
>
> Thank you for this suggestion. We initially did not include EDM results due to the substantial computational cost: generating 1M images using EDM requires 4 days with 16×H100 GPUs. While BRONet generated 2M images, these were not publicly released. We have now generated 1M EDM samples and conducted the comparison. The results are shown below:
>
> |model|Use EDM|ε=1|ε=36/255|
> |:-:|:-:|:-:|:-:|
> |L32W4096|no|40.2/21.1|55.9/40.3|
> |L32W4096|yes|41.4/21.9|57.2/41.4|
>
> These results demonstrate that training with EDM-generated data provides consistent improvements across different perturbation levels. However, we believe future work should focus on using large amounts of text-image data (i.e., apply CLIP models to Lipschitz training), rather than scaling diffusion generated data

---

> ### Author Response · Authors · 2025-11-26
> **Response to Reviewer BPtS (2/2)**
>
> ## Question 4 & 5
> > the relationship between β-Abs and min–max. In Theorem 2, is it necessary to use this R? Also, please verify whether line 751 is correct.
>
> Theorem 2 establishes that β-Abs is mathematically equivalent to MinMax with rotations when β=0.5. Specifically, MinMax(x) is equivalent to rotating x, applying β-Abs, and then rotating back. The rotation matrix R is indeed necessary to establish this equivalence. Thank you for identifying the issue in line 751—there is a typo in the proof, which we will correct in the revised version.
>
> > SOC introduces other activation functions such as Householder and GroupSort. Can the paper compare with these? Otherwise, the advantage of β-Abs is unclear.
>
>  The motivation for β-Abs is to provide a more GPU-efficient activation function while maintaining orthogonality. Prior activation functions have significant computational overhead:
>
> - MinMax: requires swapping values between tensor partitions
> - HouseHolder: requires 4 logical operations and 12 sin/cos operations
>
> In contrast, β-Abs uses only an in-place operation, resulting in substantial efficiency gains. The table below compares training speed (minutes/epoch) for LipNeXt on a 4×A100 machine:
> |Model Config |$\beta$-Abs | MinMax |HouseHolder |
> |:-:|:-:|:-:|:-:|
> |LipNeXt L32W1024| 0.18 | 0.23 | 0.79 |
> |LipNeXt L32W2048| 0.57 | 0.66 | 1.69 |
>
> β-Abs achieves approximately 10% training speedup compared to MinMax, without requiring any other architectural changes. We believe such GPU-efficiency property is preferred for model scaling up. Despite our efforts to optimize the Householder activation implementation, it remains significantly slower, making it impractical for large-scale models.
>
> ## Question 6
>
> > What does the dagger (in line 351) symbol in Table 1 represent?
>
> Thank you for catching this. This is a typo. We initially intended to use results from the original LiResNet paper and denote that the model was trained with diffusion-generated data. However, we subsequently found reproduced results from the BRONet paper but neglected to remove the symbol. We will remove this symbol in the revision.
>
> ## Question 7
> >  Why does the paper use R in Equation (8)? Is this orthogonal mapping essential and trainable?
>
> Yes, R is a learned orthogonal matrix parameter (together with M in Equation 9). We conducted ablation studies comparing learned versus fixed orthogonal matrices. Using fixed orthogonal matrices reduces the parameter count by approximately 50% (e.g., from 64M to 32M parameters). However, our experiments demonstrate that the learned version consistently outperforms the fixed version when controlling for parameter count.
>
> The orthogonal projection R is essential for the effectiveness of spatial shifting. Without R, the spatial shifting operation would consistently shift the same subset of channels across all layers, severely limiting representational capacity. The projection mechanism works as follows: R performs a learned channel permutation/mixing, while R^T reverses this transformation. This design ensures that different feature channels are shifted across different layers.
>
> ## Question 8
> > could the authors clarify its specific effect?
>
> Thank you for this insightful question. The Riemannian gradient derived in Equations 2 and 3 assumes that the optimized term X is orthogonal. However, because we employ Taylor series truncation for the matrix exponential estimation, X may deviate from orthogonality during optimization. To maintain the accuracy of our Riemannian gradient, we project X back onto the orthogonal manifold.
>
> > Why not use other orthogonal projections such as LOT or Cholesky?
>
> We use polar retraction because it finds the closest orthogonal matrix to X in the Frobenius norm sense, which is theoretically well-motivated for our optimization procedure. In principle, LOT or Cholesky decomposition could also be employed. Since X remains close to orthogonal throughout training, all three methods should yield similar outputs. We chose polar retraction for its well-established theoretical properties and geometric interpretation.
>
> ## Question 9
>
> > include AutoAttack results to verify that the certified robustness implementation is correct.
>
> Thank you for this valuable suggestion. We have verified our implementation using LipNeXt L32W2048 trained on CIFAR-10. The table below compares certified robustness accuracy (CRA) with AutoAttack robustness accuracy (including apgd-ce, apgd-t, fab-t, and square attacks):
>
> |metric | $\varepsilon=36/255$| $\varepsilon=72/255$|$\varepsilon=108/255$|
> |:-:|:-:|:-:|:-:|
> |CRA| 73.2 |58.8 |43.3|
> |Auto-attack Acc| 79.6| 74.2 | 68.0 |
>
> As expected, the AutoAttack accuracy is higher than the certified accuracy across all perturbation radii, which validates that our certified robustness implementation is correct.
>
> ## Question 10
> > the following reference may be missing
>
> Thank you for highlighting this omission. We will discuss this work in our related work.

---

### Meta-Review · Area_Chair_VFQK · 2026-01-09

**Summary:**

This paper proposes LipNeXt, a new architecture and training procedure for large-scale 1-Lipschitz networks.
All three reviewers agree that the paper is technically sound, well written, and presents a novel and rigorous approach to scaling Lipschitz-based certified robustness to very large models.

The reviewers' main concerns include the fairness of comparisons due to model size differences, missing or incomplete baselines and references, and high computational and memory costs. The authors’ rebuttal substantially reduced these concerns by adding missing experiments, clarifying definitions, and justifying design choices. After the rebuttal, no major technical flaws remain. The remaining weaknesses are mainly about resource intensity and accessibility, not correctness or novelty. This led to a positive overall assessment.

**Reviewer Concerns:**

Addressed by the rebuttal
- Missing baselines and fairness of comparison (Reviewers BPtS, cWBh, ZYaL) → Addressed by adding size-matched results (LipNeXt L32W1024), scaling experiments for LiResNet and BRONet, and explaining why some baselines fail to scale.
- Clarification of CRA definition and norm choice (Reviewer cWBh) → Formally defined CRA, clarified ℓ₂ norm, and distinguished certified from empirical robustness.
- Role and benefit of β-Abs (Reviewers BPtS, cWBh) → Clarified theoretical equivalence to MinMax, efficiency benefits, and its role as a computational improvement rather than the main performance driver.
- Justification of orthogonal projections, spatial shift, and polar retraction** (Reviewers BPtS, ZYaL) → Provided explanations, ablations, and theoretical motivation.
- Implementation correctness of certification (Reviewer BPtS) → Addressed by reporting AutoAttack comparisons.
- Resource and reproducibility concerns** (Reviewer ZYaL) → Mitigated by reporting results on smaller hardware and explaining memory optimization via ZeRO.
- Missing references and related work gaps (Reviewers BPtS, ZYaL) → Acknowledged and promised to add.

⠀Still outstanding (minor)
* The approach remains computationally expensive and parameter-heavy, limiting accessibility.
* Some architectural choices may feel heavyweight compared to potentially more parameter-efficient alternatives, but this is more a limitation of the approach than a flaw.

No unresolved concerns question the validity, correctness, or novelty of the work.

**Reviewer Scores:**

Below is an estimate of how each reviewer’s score would likely change after considering the rebuttal and discussion.

Reviewer BPtS
Original score: 6.
Many of their specific questions (baselines, EDM, β-Abs, AutoAttack, typos) were directly addressed.
Likely updated score: remains 6.

Reviewer cWBh
Original score: 6, with low confidence.
The rebuttal clarified CRA, norm choice, β-Abs motivation, and provided the requested smaller model.
Likely updated score: remains 6.

Reviewer ZYaL
Original score: 8, high confidence.
The concerns were explicitly addressed, and the reviewer confirmed satisfaction in a follow-up comment.
Likely updated score: remains 8.

Overall, after rebuttal and discussion, the consensus would shift from “borderline accept” to a solid accept with no remaining blocking issues.

---

### Decision · Program_Chairs · 2026-01-26

Accept (Poster)